# OpenReview forum: "Language Model Alignment in Multilingual Trolley Problems"
_ICLR.cc/2025/Conference — ICLR 2025 Spotlight_

### Official Review · Reviewer_DzVX · 2024-11-02

**Soundness:** 3
**Presentation:** 4
**Contribution:** 2
**Rating:** 8
**Confidence:** 4

**Summary:**

The paper introduces a dataset of Multilingual Trolley Problems. The typical task frames a hypothetical situation in which a self-driving car's brakes are broken and it hits one of two specified characters.   The dataset covers 107 languages, six moral dimensions, and 18 characters. The paper prompts 19 LLMs to investigate the alignment between LLM choices in the trolley problem and human preferences collected in prior studies. The paper finds that LLMs vary in alignment with human preferences and exhibit different behavior when prompted in various languages.

**Strengths:**

* The paper presents a novel dataset designed to evaluate alignment in a multilingual setup. The dataset has a solid design with parametric variation and covers 109 languages, 6 moral dimensions, and 18 characters. The data is translated using a machine translation engine, which is suitable for this task since it is not tied to linguistic subtleties.

* The experimental protocol appears robust. LLMs are prompted with one of the problems, and preferences are extracted from their responses. A misalignment score is then computed to capture differences between LLM and human preferences.

* The results discussion is organized into five sections aligned with specific research questions, making it easy to follow and understand the key findings. These findings address overall alignment, LLM performance across six moral dimensions, language variation, performance in low-resource languages, and LLM robustness to prompt paraphrasing.

**Weaknesses:**

* Although the paper offers a detailed analysis of LLM performance across several research questions, it lacks an exploration of potential reasons behind the observed phenomena. While such explanations might be speculative given the limited information on LLM training, a discussion on why models from the same family (e.g., Llama 3) exhibit differing behaviors could enhance the paper.

* The paper implicitly assumes that LLMs should align with country-level or language-level human preferences. However, these preferences may carry biases and could be unfair to marginalized groups. Conversely, the least aligned model, GPT-4o Mini, could be viewed as one that lacks language-specific biases and instead follows a standardized behavior pattern. Exploring this topic further in the paper could add valuable insights into the potential trade-offs between alignment and bias in multilingual settings.

**Questions:**

* What factors might account for differences in behavior between models?
* Can you elaborate more on the post-processing procedure in L276?
* Did you observe any effect of position bias in the option order in L233?

---

> ### Author Response · Authors · 2024-11-23
>
> Thank you for your positive feedback that our paper “presents a novel dataset”, “has a solid design with parametric variation and covers 109 languages, 6 moral dimensions, and 18 characters”, has “robust” “experimental protocol”, and “The results discussion” is “aligned with specific research questions, making it easy to follow and understand the key findings.”
>
> > The paper implicitly assumes that LLMs should align with country-level or language-level human preferences. However, these preferences may carry biases and could be unfair to marginalized groups.
>
> This is a key point, and we have updated the manuscript to directly address this assumption. Simplifying a bit – one might consider two possible goals for LLM alignment. One could aim to align as closely as possible with human preferences (and, in the case of pluralistic values [1], personalize to the user) or align with a universal principle such as fairness or impartiality for all users. These two goals can come into conflict, but it is not our focus on resolving this conflict. Instead, our focus is on measurement and evaluation – to first answer the question of how do current LLMs align with diverse populations. We have added the following text to  the Ethics Statement of our manuscript, together with clarification in Section 3.1:
>
> “The Moral Machine dataset (Awad et al. 2018) is a _descriptive_ measure of human responses to moral dilemmas. It cannot (and should not) _prescribe_ how the system should act. However, because these systems are trained and fine-tuned with large-scale cross-cultural data, they may inherit culturally specific biases. Our focus here is to both characterize these biases and study to what extent input in different languages leads to an LLM reflecting the moral values consistent with the speakers of that language. From this perspective, a higher alignment score for a particular culture may or may not be desirable. While some point to reflecting human values in AI systems as a solution to the alignment problem, our results suggest that due to the plurality of moral values expressed, successful alignment must ultimately depend on the intent of the system developers to avoid perpetuating human preferences that are considered harmful, partial, or parochial.”
>
> [1] Sorensen et al., Position: A Roadmap to Pluralistic Alignment, 2024.
>
> > the least aligned model, GPT-4o Mini, could be viewed as one that lacks language-specific biases and instead follows a standardized behavior pattern. Exploring this topic further in the paper could add valuable insights into the potential trade-offs between alignment and bias in multilingual settings.
>
> This is a very interesting hypothesis! Our RQ1 addresses the overall alignment of the models, and RQ3 and RQ4 report on the results for different families of languages. As reported in Section 5.4, there was no language bias for Llama 3.1 70B, with a correlation close to 0. In our appendix, we show the language bias for all the models in Table 10, and find an overall close-to-zero Pearson correlation between the misalignment score and the number of speakers of each language, again showing little bias across different levels of misalignment scores.
>
>
>
> > Can you elaborate more on the post-processing procedure in L276?
>
> Since the raw response from LLMs is free-form text, we parse every response into the given choices of the trolley problems. For example, for the question in Figure 1 on “a boy” vs “an elderly man”, LLMs may provide answers such as “It should save the boy” or “It should save the younger.” To process the free text answers into the given choices presented in our trolley problem prompts, we first run exact string matching of the choices (e.g., matching the exact word “boy” with the first option), and then as a fallback, in case a good match is not found, we prompt an LLM to parse the answer (e.g., parsing the response “the younger” as the first option of saving the boy too) using Llama 3.1 8B.
>
> > Did you observe any effect of position bias in the option order in L233?
>
> In pilot studies, we studied this exact question to make sure our scenario descriptions are robust. We have added these results to Appendix E.5 since they may prove useful to other researchers. Briefly, we conducted a perturbation experiment where we swapped the order of the two options and recorded the number of choice reversals. We report the consistency rate as the percentage of times when the models’ answers stay the same. We find that LLMs are robust against position bias, as most models have 97-99.5% consistency rates in Table 13.
>
>
> ---
>
> We hope our answers address your concerns and can clarify the value of our work. In case you need any followup information, we are more than happy to provide more details!

---

### Official Review · Reviewer_sLsu · 2024-11-03

**Soundness:** 3
**Presentation:** 4
**Contribution:** 3
**Rating:** 8
**Confidence:** 4

**Summary:**

The paper investigates how large language models (LLMs) align with human moral preferences across various languages and cultures. It introduces the MULTITP dataset, based on the Moral Machine experiment, which collected over 40 million human judgments globally. MULTITP includes 97,520 translated moral dilemma scenarios in 107 languages, allowing for a systematic evaluation of LLMs' moral reasoning across linguistic contexts.

The study assesses 19 LLMs across six moral dimensions: species, gender, fitness, social status, age, and the number of lives involved. The findings show that while most LLMs do not closely align with human moral judgments, they display similar levels of alignment across both major and minor languages, countering assumptions about biases related to language resources.

Key contributions include providing a comprehensive dataset for multilingual moral alignment, developing methods to analyze variations in moral dilemmas, and creating frameworks for assessing cross-cultural alignment. The paper underscores limitations in LLMs' ability to match human moral reasoning while emphasizing the need to integrate diverse cultural and linguistic perspectives into AI ethics research. This work serves as both a technical guide for measuring alignment and a reminder of the complexities involved in cross-cultural moral decision-making in AI.

**Strengths:**

**Originality**: The paper introduces an innovative approach by adapting the Moral Machine framework to evaluate LLMs across languages, creating the unique MULTITP dataset for consistent cross-lingual moral dilemma evaluation, and developing novel metrics for measuring alignment.

**Quality**: The study demonstrates rigorous methodology through a comprehensive evaluation of 19 LLMs, systematic moral dimension variations, robust prompt paraphrasing tests, and careful statistical analysis with transparency in acknowledging limitations.

**Clarity**: The paper is well-organized with clear research questions, strong visual aids, detailed methodological documentation, and effective use of examples to communicate complex findings.

**Significance**: The work contributes significantly to understanding cross-cultural AI alignment, challenges assumptions about language inequality, and provides essential tools for future research, with implications for creating culturally aware and fair multilingual AI systems.

**Weaknesses:**

**Dataset Transformation**: Although the dataset is large and comprehensive, it is adapted from the existing *Moral Machine* experiment, raising questions about its novelty. The paper would benefit from clarifying what unique modifications or contributions were made to enhance the dataset beyond its scale and multilingual coverage.

**Translation Quality**: Maintaining high translation quality across a vast number of questions, especially in low-resource languages, is challenging. The reliance on *googletrans* (the non-commercial open-source tool) may compromise accuracy.

**Alignment Claims**: The study claims that LLMs generally do not align well with human preferences, but it aggregates data from all cultural contexts, potentially obscuring the nuances of cultural alignment. Expecting models to align with an average of global human preferences may be unrealistic or inappropriate, given the significant variability in cultural moral standards. Discussing how LLMs perform within specific cultural subsets might provide more nuanced insights.

**Practical Relevance for LLM Development**: While the study offers valuable theoretical insights, its direct application to real-world LLM development is not clear. Demonstrating how these findings could inform practical strategies for LLM fine-tuning and alignment would make the paper more relevant for AI practitioners.

**Use of Binary Moral Dilemmas**: The paper’s reliance on binary moral dilemmas, such as trolley problems, provides a structured evaluation but oversimplifies the complexities of real-world ethical decision-making. Real-life moral situations are rarely clear-cut, and binary choices may not capture the depth of human reasoning. Including more complex and context-sensitive scenarios could enhance the study’s applicability and realism.

**Interpretation of Results**: Although the paper includes multiple languages, it does not explore how cultural nuances influence LLM responses. Cultural context significantly shapes moral judgments, and not addressing these variations may miss key insights into LLM behaviour in diverse moral frameworks. Analyzing language-specific and culture-specific responses would enrich the findings and offer a deeper cross-cultural understanding.

**Questions:**

- Could you clarify why a score of 0.6 in RQ1 is considered close alignment to human preference? While I understand that the scores can be used to rank LLM alignment, I am curious whether the absolute value holds inherent significance. In other words, what justifies considering a 0.6 score as an indicator of close alignment between LLMs and human preferences?
- Could you provide more details on the manual review process for translations, including the number of samples reviewed, the criteria used for evaluation, and any systematic issues or errors that were found during this process?
- What steps or analysis would be necessary to evaluate LLM performance within specific cultural subsets, and did your findings reveal any trends where certain models align more closely with specific **cultural norms**?
- Did you observe any significant differences in alignment or misalignment patterns between major and minor languages? How might these insights inform future strategies for training and fine-tuning language models?

**Details Of Ethics Concerns:**

This paper releases a dataset that might involve ethical concerns, such as bias or fairness issues in different cultures. The study includes human subjects.

---

> ### Author Response · Authors · 2024-11-23
> **Response to Reviewer sLsu (1/n)**
>
> We appreciate your positive feedback that our work “introduces an innovative approach” to create the “unique MULTITP dataset,” which is “a comprehensive dataset for multilingual moral alignment,” and presents “novel metrics for measuring alignment.” Thanks for further commenting that our work “demonstrates rigorous methodology” and “careful statistical analysis with transparency in acknowledging limitations,” and “contributes significantly to understanding cross-cultural AI alignment.”
>
> > The paper would benefit from clarifying what unique modifications or contributions were made to enhance the dataset beyond its scale and multilingual coverage.
>
> Thank you for the suggestion! The key modifications we have made are as follows: First, to counteract the tendency of LLMs to avoid providing a clear answer, we employ the token-forcing method to force the LLMs to make a binary choice. Second, we expand the original natural language data from English into 107 languages in total. The dataset now covers a wide range of languages including those with many speakers (high-resource) and those with few (low-resource). Third, the stimuli are now formatted to be compatible with LLMs. We reduce arbitrary bias by removing keywords such as “swerve” and presenting the two choices using the bullet points to reduce position bias for the first option over the second. These modifications enable the research community to leverage the Moral Machine dataset to study and evaluate LLM pluralistic alignment.
>
> We have edited the text in Section 3.2: “Setup for LLM Testing” to clarify these contributions.
>
> > The reliance on googletrans (the non-commercial open-source tool) may compromise accuracy.
> > Could you provide more details on the manual review process for translations, including the number of samples reviewed, the criteria used for evaluation, and any systematic issues or errors that were found during this process?
>
> Yes, we can provide the details of our translation quality check. Specifically, we have conducted a new study on Amazon Mechanical Turk (MTurk) to evaluate the translation quality using annotators who identified as being proficient in any of our given languages in addition to English. **TLDR: Manual annotation shows that translation quality was high across languages.** The following details are in the Appendix D of the manuscript:
>
> Details of the annotation task: We designed a task on Amazon Mechanical Turk (Mturk) where annotators rated how accurately our translations reflected the meanings of the original English prompts. Ratings were provided on a 5-point scale, from non-sense translation to accurate translation. First, annotators were prompted to select the language they have the greatest proficiency in from a full list of 106 languages (excluding English). If they were not proficient in any of the listed languages, they did not proceed. Then, after selecting a language, annotators read 25 translation pairs (English sentences or words paired with their translation in the selected language). Instructions and rating scales were adapted from [1, 2] as follows:
>
>    > Please rate how accurately the translation conveys the meaning of the source text:
>    > Source English Sentence/Text: {source}
>    > Translation: {translation}
>    > - None (Score=1): No meaning of the original text is conveyed.
>    > - Little (Score=2): Only a small portion of the meaning is conveyed.
>    > - Much (Score=3): A substantial amount of the meaning is conveyed.
>    > - Most (Score=4): Most of the meaning is conveyed accurately.
>    > - All (Score=5): The full meaning is conveyed accurately.
>
> We kept the Amazon Mechanical Turk task open for a span of three days, and collected a total of 169 annotation responses covering 44 languages. The results of the evaluation demonstrate strong translation quality. Specifically, 88.6% of the evaluated languages achieved an average score of at least 3.0 out of 5 (i.e., conveying a substantial portion of the original meaning), and no languages received a mean score below 2.0.
>
> The standard deviation of scores across languages ranged from 0.5 to 1.3, reflecting relatively consistent ratings among annotators. Further details on the evaluation setup and results are included in Appendix D.
>
> [1] Lavie, Alon, Evaluating the Output of Machine Translation Systems, 2011
>
> [2] Goto et al., Crowdsourcing for Evaluating Machine Translation Quality, 2014

---

> ### Author Response · Authors · 2024-11-23
> **Response to Reviewer sLsu (2/n)**
>
> > [the study] aggregates data from all cultural contexts, potentially obscuring the nuances of cultural alignment. Expecting models to align with an average of global human preferences may be unrealistic or inappropriate, given the significant variability in cultural moral standards […] Analyzing language-specific and culture-specific responses would enrich the findings and offer a deeper cross-cultural understanding.
>
> Thank you for the opportunity to let us clarify this. Our RQ3 and RQ4 directly address language-specific preference variations. Specifically, the results from RQ3 show significant variance across different languages (with the relatively high language sensitivity scores reported in Table 3), and RQ4 further shows how moral preferences vary systematically across the four different language clusters (Figure 4). In the Appendix E.4 (Figure 8), we visualize a world map that shows country-specific alignment patterns. These visualizations also help us understand cultural subsets and support a key conclusion of RQ4: we do not find a distinct alignment bias toward low-resource languages.
>
> > What steps or analysis would be necessary to evaluate LLM performance within specific cultural subsets, and did your findings reveal any trends where certain models align more closely with specific cultural norms?
>
> As introduced in our answer for the previous question, our RQ3 and RQ4 directly address language-specific preference variations. In addition, for the rebuttal, we added a new analysis of the distribution of moral preferences across three major geographic regions in Appendix E.4.1, covering the regional preferences of the Global East, West, and South countries. This enables us to investigate potential cultural variations in moral decision-making. We plot the distribution of moral preferences across the three regional clusters in Appendix E.4.1.
>
> > While the study offers valuable theoretical insights, its direct application to real-world LLM development is not clear.
>
> By analyzing misalignment across moral dimensions and languages, practitioners can identify specific biases, such as over-prioritizing certain demographic attributes, and address these through targeted fine-tuning as needed. For example, incorporating multilingual moral judgments into reward models may influence ethical reasoning capabilities across diverse linguistic contexts and give developers better control and visibility into the populations they are aligning towards. Furthermore, our structured moral dilemmas offer a template for testing model behavior post-alignment, ensuring consistency and robustness. These practical methodologies pave the way for deploying LLMs with improved and more transparent ethical alignment, directly benefiting the users of AI systems operating across culturally diverse settings.
>
> > Real-life moral situations are rarely clear-cut, and binary choices may not capture the depth of human reasoning. Including more complex and context-sensitive scenarios could enhance the study’s applicability and realism.
>
> We think this is an important area for future work and discuss it in the last paragraph of our Ethics Statement section in Appendix. We look forward to future follow-ups with richer and more diverse scenarios capable of capturing more nuance in human moral decision-making and judgment.
>
> > Could you clarify why a score of 0.6 in RQ1 is considered close alignment to human preference?
>
> We are happy to clarify! We do not intend to say that misalignment scores of 0.6 are to be interpreted as having close alignment with human preferences. Our goal in calling out the number 0.6 is that this was approximately the lowest misalignment we observed across the 19 LLMs tested, so we tried to give some intuition to the reader about the relationship between the global misalignment and the components that go into its calculation. We make clear in Section 5.1 Results that 0.6 is not a hard cutoff for close alignment with human preferences and that even for the best models, much work remains to be done.

---

> ### Author Response · Authors · 2024-11-23
> **Response to Reviewer sLsu (3/n)**
>
> > Did you observe any significant differences in alignment or misalignment patterns between major and minor languages?
>
> Yes! One of the most surprising results in our manuscript is shown in RQ4 (Section 5.4). There, we show that most models lack a significant correlation between the alignment scores and the major/minor nature of the languages – correlations are close to zero. As a clarification, in the paper we use a nomenclature high-resource and low-resource for major and minor languages, respectively. For example, in Llama 3.1 70B, the misalignment scores for the top five most spoken languages are: Chinese 0.38, Hindi 0.51, English 0.58, Spanish 0.68, and Arabic 0.54. This distribution of correlations is about the same as in less spoken languages among our dataset: Bosnian 0.54, Luxembourgish 0.37, Icelandic 0.58, Maltese 0.57, Malayalam 0.54, and Catalan 0.59. We also visualize a misalignment world map in Appendix D.3. The maps again show a lack of a strong correspondence between the approximate development level of a country and its alignment score.
>
> ---
>
> We hope our answers address your concerns and can clarify the value of our work. In case you need any followup information, we are more than happy to provide more details!

---

> > ### Comment · Reviewer_sLsu · 2024-11-24
> >
> > Hi, the rebuttal is satisfactory, and I have raised my score. Please make the necessary revisions to the paper accordingly.

---

> > > ### Author Response · Authors · 2024-11-25
> > >
> > > Thank you for taking the time to review our rebuttal and for raising your score. We truly appreciate your understanding and thoughtful feedback. We’ve carefully incorporated all the promised revisions into the updated PDF submission.

---

### Official Review · Reviewer_JShf · 2024-11-04

**Soundness:** 3
**Presentation:** 4
**Contribution:** 3
**Rating:** 8
**Confidence:** 3

**Summary:**

The study investigates how well LLMs align with human moral preferences in multilingual settings, particularly through the lens of trolley problem scenarios. The authors introduce a new dataset called MultiTP, which is generated by filling templates with different variables and translating them into multiple languages using Google Translate. They conducted experiments and concluded that LLMs do not strongly align with human preferences but did not show significant alignment inequality between high-resource and low-resource languages.

**Strengths:**

The paper is well-structured and clearly written. It addresses an important new challenge in aligning LLMs' responses with human preferences in moral decision-making. The authors conducted detailed experiments and provided valuable insights. The MultiTP dataset, which spans numerous languages, could be highly useful for future studies on aligning model behavior with human ethics.

**Weaknesses:**

One of the main concerns is the justification for aligning LLMs with demographic distributions and human preferences. This approach might introduce biases into the models. For example, in certain cultures where laws or social norms may place different values on individuals (e.g., men being valued more than women in some religions), aligning LLMs with such preferences could reinforce harmful biases. The examples provided in Figure 1 illustrate cultural biases related to age, which could extend to other sensitive attributes. To address these issues, the authors should clarify the practical use cases for aligning models in such contexts and explain how alignment can be distinguished from bias, as well as how fairness is maintained.

Another concern is the reliability of using Google Translate for generating the dataset in various languages. Although the authors reviewed a subset of translations for major languages, translation quality tends to decrease for less common languages, which could impact the consistency of the dataset and the validity of the findings. For example, in the Persian translation, Google Translate incorrectly rendered the phrase "it [the self-driving car] should save" by translating "save" as "صرفه جویی کردن" (meaning "to economize" which is a verb usually used for resources, like water) instead of the correct "نجات دادن' (meaning "to rescue/save life"). This confusion occurs despite Persian being a widely-spoken language with substantial digital presence. Such semantic errors in a relatively well-resourced language raise serious concerns about the translation quality for truly low-resource languages, where training data is far more limited.

Lastly, the study does not account for performance differences between LLMs across various languages. Models like GPT-4 and Llama 2 are expected to show significant performance variations in low-resource languages, potentially affecting the reliability of the results in experiments such as RQ1 and RQ4.

**Questions:**

Q1: Could noise factors, such as the performance of Google Translate in translating prompts or the effectiveness of LLMs in low-resource languages, impact the alignment of model preferences and potentially introduce discrepancies or inconsistencies in multilingual outputs?

Q2: How would you handle cases where human preferences conflict with principles of fairness and equality?

---

> ### Author Response · Authors · 2024-11-23
> **Response to Reviewer JShf (1/n)**
>
> Thank you for your positive feedback that our paper is “well-structured and clearly written,” “addresses an important new challenge”, and “provided valuable insights” with “detailed experiments”. We will address your potential concerns in the following.
>
> > Although the authors reviewed a subset of translations for major languages, translation quality tends to decrease for less common languages, which could impact the consistency of the dataset and the validity of the findings.
>
> Thank you for double-checking the translation quality. Some translation tasks might be challenging as different translations map to the same English word, as in the Persion example you mentioned. To quantitatively address the translation quality, we have conducted a new study on Amazon Mechanical Turk (MTurk) to evaluate the translation quality using annotators who identified as being proficient in any of our given languages in addition to English. **TLDR: Manual annotation shows that translation quality was high across languages.** The following details have been added to Appendix D of the manuscript:
>
> Details of the annotation task: We designed a task on Amazon Mechanical Turk (Mturk) where annotators rated how accurately our translations reflected the meanings of the original English prompts. Ratings were provided on a 5-point scale, from non-sense translation to accurate translation. First, annotators were prompted to select the language they have the greatest proficiency in from a full list of 106 languages (excluding English). If they were not proficient in any of the listed languages, they did not proceed. Then, after selecting a language, annotators read 25 translation pairs (English sentences or words paired with their translation in the selected language). Instructions and rating scales were adapted from [1, 2] as follows:
>
>    > Please rate how accurately the translation conveys the meaning of the source text:
>    > Source English Sentence/Text: {source}
>    > Translation: {translation}
>    > - None (Score=1): No meaning of the original text is conveyed.
>    > - Little (Score=2): Only a small portion of the meaning is conveyed.
>    > - Much (Score=3): A substantial amount of the meaning is conveyed.
>    > - Most (Score=4): Most of the meaning is conveyed accurately.
>    > - All (Score=5): The full meaning is conveyed accurately.
>
> We kept the Amazon Mechanical Turk task open for a span of three days, and collected a total of 169 annotation responses covering 44 languages. The results of the evaluation demonstrate strong translation quality. Specifically, 88.6% of the evaluated languages achieved an average score of at least 3.0 out of 5 (i.e., conveying a substantial portion of the original meaning), and no languages received a mean score below 2.0.
>
> The standard deviation of scores across languages ranged from 0.5 to 1.3, reflecting relatively consistent ratings among annotators. Further details on the evaluation setup and results are included in Appendix D of our updated paper PDF.
>
> [1] Lavie, Alon, Evaluating the Output of Machine Translation Systems, 2011
>
> [2] Goto et al., Crowdsourcing for Evaluating Machine Translation Quality, 2014

---

> ### Author Response · Authors · 2024-11-23
> **Response to Reviewer JShf (2/n)**
>
> > For example, in certain cultures where laws or social norms may place different values on individuals (e.g., men being valued more than women in some religions), aligning LLMs with such preferences could reinforce harmful biases. [...] How would you handle cases where human preferences conflict with principles of fairness and equality?
>
> Thank you for pointing out this important and nuanced point. We have added the following text to the Ethics Statement of our manuscript:
>
> _“The Moral Machine dataset (Awad et al. 2018) is a _descriptive_ measure of human responses to moral dilemmas. It cannot (and should not) _prescribe_ how the system should act. However, because these systems are trained and fine-tuned with large-scale cross-cultural data, they may inherit culturally specific biases. Our focus here is to both characterize these biases and study to what extent input in different languages leads to an LLM reflecting the moral values consistent with the speakers of that language. From this perspective, a higher alignment score for a particular culture may or may not be desirable [3, 4]. While some point to reflecting human values in AI systems as a solution to the alignment problem (i.e., pluralistic alignment [5]), our results suggest that due to the plurality of moral values expressed, successful alignment must ultimately depend on the intent of the system developers to avoid perpetuating human preferences that are considered harmful, partial, or parochial [5, 6].”_
>
> To make the spirit clear, we also added a clarification sentence earlier in the paper when introducing the dataset in Line 175 in Section 3.1.
>
> [3] Bommasani et al., On the Opportunities and Risks of Foundation Models, 2021.
>
> [4] Ethayarajh et al., The Authenticity Gap in Human Evaluation, 2022.
>
> [5] Sorensen et al., Position: A Roadmap to Pluralistic Alignment, 2024.
>
> [6] Santurkar et al., Whose Opinions Do Language Models Reflect? 2023.
>
> ---
>
> We hope our answers address your concerns and can clarify the value of our work. In case you need any followup information, we are more than happy to provide more details!

---

> > ### Comment · Reviewer_JShf · 2024-11-25
> >
> > Thank you for your comprehensive and detailed response. I appreciate the effort to clarify the study's intentions and the addition of the statement to the Ethics section, which indeed enhances the transparency and contextualization of your work. However, I remain somewhat skeptical about the claim of achieving high-quality translations based on the Mechanical Turk evaluation. I believe further details and transparency regarding the statistical results of the experiment, such as the distribution of scores per language and more granular insights into the annotation process, would strengthen the validity of your conclusions.
> >
> > That said, I find the overall contributions of your paper compelling and am inclined toward acceptance. Accordingly, I have increased my score.

---

> > > ### Author Response · Authors · 2024-11-25
> > >
> > > Thank you for your thoughtful feedback and for recognizing our work's contributions. We sincerely appreciate your constructive suggestions and are grateful for your increased score!
> > >
> > > As an extension of Table 7, we present the detailed distribution of scores below:
> > >
> > > |Language|Language Code|Averaged Scores|# Responses|% of Score 5|% of Score 4|% of Score 3|% of Score 2|% of Score 1|
> > > |--------|-------------|---------------|-----------|------------|------------|------------|------------|------------|
> > > |Southern Sotho|st|5.00±0.00|1|100.0|0.0|0.0|0.0|0.0|
> > > |Danish|da|5.00±0.00|1|100.0|0.0|0.0|0.0|0.0|
> > > |Bosnian|bs|4.91±0.28|1|91.3|8.7|0.0|0.0|0.0|
> > > |Turkish|tr|4.81±0.43|3|82.6|15.9|1.4|0.0|0.0|
> > > |Russian|ru|4.78±0.41|2|78.3|21.7|0.0|0.0|0.0|
> > > |Chinese (Traditional)|zh-tw|4.76±0.87|9|90.8|2.9|1.0|4.3|0.0|
> > > |Portuguese|pt|4.72±0.70|6|81.9|10.9|5.8|0.0|1.4|
> > > |German|de|4.69±0.70|6|81.2|8.0|9.4|1.4|0.0|
> > > |Tagalog|tl|4.68±0.61|4|72.8|25.0|1.1|0.0|1.1|
> > > |Swahili|sw|4.68±1.11|3|91.3|0.0|2.9|0.0|4.3|
> > > |Chinese (Simplified)|zh-cn|4.63±0.98|5|84.3|5.2|2.6|7.0|0.0|
> > > |Dutch|nl|4.61±0.71|1|73.9|13.0|13.0|0.0|0.0|
> > > |Korean|ko|4.59±1.13|3|81.2|10.1|4.3|1.4|0.0|
> > > |Bulgarian|bg|4.59±1.31|2|87.0|4.3|2.2|2.2|0.0|
> > > |Czech|cs|4.57±1.04|2|80.4|8.7|2.2|4.3|4.3|
> > > |English|en|4.53±0.73|7|60.9|34.8|3.1|0.6|0.0|
> > > |Malagasy|mg|4.52±1.12|5|79.1|7.8|6.1|1.7|4.3|
> > > |Vietnamese|vi|4.52±0.50|3|52.2|47.8|0.0|0.0|0.0|
> > > |Spanish|es|4.49±0.96|16|66.6|23.6|6.2|1.9|0.3|
> > > |Romanian|ro|4.48±0.83|2|65.2|21.7|8.7|4.3|0.0|
> > > |Belarusian|be|4.48±0.50|1|47.8|52.2|0.0|0.0|0.0|
> > > |Italian|it|4.47±1.00|5|72.2|9.6|13.9|3.5|0.0|
> > > |Japanese|ja|4.45±1.01|5|67.8|19.1|7.0|4.3|0.9|
> > > |Afrikaans|af|4.33±0.55|2|37.0|58.7|4.3|0.0|0.0|
> > > |Malay|ms|4.30±0.95|1|60.9|13.0|21.7|4.3|0.0|
> > > |Telugu|te|4.30±1.07|3|62.3|18.8|7.2|10.1|1.4|
> > > |Yoruba|yo|4.29±1.25|3|59.4|27.5|7.2|0.0|2.9|
> > > |Hungarian|hu|4.26±1.26|2|73.9|0.0|4.3|21.7|0.0|
> > > |Malayalam|ml|4.10±1.43|5|60.9|8.7|22.6|3.5|0.0|
> > > |Tamil|ta|4.06±1.14|9|46.4|30.4|10.6|7.7|4.8|
> > > |Bengali|bn|4.03±1.14|3|30.4|59.4|2.9|2.9|1.4|
> > > |French|fr|4.03±0.98|3|46.4|13.0|37.7|2.9|0.0|
> > > |Hindi|hi|3.94±1.32|11|40.7|34.0|15.0|5.9|0.8|
> > > |Irish|ga|3.83±1.17|2|37.0|19.6|39.1|2.2|0.0|
> > > |Gujarati|gu|3.83±1.86|1|69.6|0.0|0.0|13.0|13.0|
> > > |Urdu|ur|3.64±1.43|4|37.0|17.4|31.5|9.8|0.0|
> > > |Hmong|hmn|3.61±1.86|1|30.4|52.2|4.3|0.0|0.0|
> > > |Arabic|ar|3.58±1.39|3|33.3|31.9|0.0|31.9|1.4|
> > > |Armenian|hy|3.28±1.27|14|18.0|29.8|25.8|17.4|7.8|
> > > |Finnish|fi|2.70±1.40|1|4.3|21.7|39.1|26.1|0.0|
> > > |Latin|la|2.50±1.31|4|6.5|18.5|18.5|38.0|15.2|
> > > |Albanian|sq|2.30±2.11|2|2.2|47.8|8.7|13.0|2.2|
> > > |Polish|pl|2.26±0.53|1|0.0|4.3|17.4|78.3|0.0|
> > > |Amharic|am|2.00±0.00|1|0.0|0.0|0.0|100.0|0.0|

---

> > > ### Author Response · Authors · 2024-11-25
> > >
> > > > Lastly, the study does not account for performance differences between LLMs across various languages.
> > >
> > > We address this concern by evaluating the tested LLMs' consistency across languages using three types of analyses: (1) **response eligibility across languages to our MultiTP questions**, (2) **robustness against option order bias**, and (3) **consistency under paraphrase perturbations**.
> > >
> > > First, we manually evaluated their ability to produce eligible responses across languages. We assessed whether their responses (back-translated into English) directly addressed the question without misinterpretation or introducing unrelated queries. For this evaluation, we randomly sampled 100 responses per language and categorized the results based on the percentage of eligible answers. Below are the findings for GPT-4o-mini and Llama 3.1 70B Instruct:
> > >
> > > GPT-4o-mini:
> > >
> > > | **Percentage of Eligible Responses** | **Proportion of Languages** | **Example Languages** |
> > > |--------------------------------------|-----------------------------|------------------------|
> > > | 90%-100%                             | 75.5%                       | sl (Slovenian), pl (Polish), nl (Dutch), jw (Javanese), fy (Western Frisian)   |
> > > | 70%-90%                              | 20.8%                       | mn (Mongolian), pt (Portuguese), ur (Urdu), sn (Shona), ug (Uyghur)   |
> > > | Below 70%                            | 3.8%                        | lo (Lao), mg (Malagasy), sd (Sindhi), zu (Zulu)       |
> > >
> > >
> > > Llama 3.1 70B Instruct:
> > >
> > > | **Percentage of Eligible Responses** | **Proportion of Languages** | **Example Languages** |
> > > |--------------------------------------|-----------------------------|------------------------|
> > > | 90%-100%                             | 67.0%                       | de (German), mr (Marathi), mk (Macedonian), ca (Catalan), cs (Czech)   |
> > > | 70%-90%                              | 23.6%                       | yi (Yiddish), mn (Mongolian), ar (Arabic), iw (Modern Hebrew), sw (Swahili)    |
> > > | Below 70%                            | 9.4%                        | lo (Lao), eu (Basque), ha (Hausa), am (Amharic), my (Burmese)   |
> > >
> > >
> > > As shown in the tables, the majority of languages achieve eligibility rates above 70% for both models. This demonstrates that performance differences across languages are generally limited and do not significantly impact the validity of our results.
> > >
> > > Second, we conducted pilot experiments to test for option order bias. We studied this exact question to make sure our scenario descriptions are robust. We have added these results to Appendix E.5 since they may prove useful to other researchers. Briefly, we conducted a perturbation experiment where we swapped the order of the two options and recorded the number of choice reversals. We report the consistency rate as the percentage of times when the models’ answers stay the same. We find that LLMs are robust against position bias, as most models have 97-99.5% consistency rates in Table 13.
> > >
> > > Third, as described in Section 5.5, we evaluated the models' consistency when questions were paraphrased. Both Llama 3.1 70B and Llama 3.1 8B demonstrated moderate to substantial consistency scores under this perturbation.
> > >
> > > The results from the three analyses confirm that performance differences across languages are limited and do not significantly interfere with the reliability of our findings. We would be happy to address any additional questions from the reviewer.

---

### Official Review · Reviewer_Q4Jg · 2024-11-04

**Soundness:** 2
**Presentation:** 4
**Contribution:** 2
**Rating:** 5
**Confidence:** 5

**Summary:**

The paper examines the moral judgments of large language models (LLMs) in a multilingual context, aiming to analyze how these models respond to ethical dilemmas. It specifically focuses on the moral dimensions involved in decision-making processes, utilizing a dataset that encompasses a variety of cultural perspectives. The authors investigate how LLMs interpret complex moral scenarios, employing methodologies such as statistical analyses and clustering techniques to evaluate the models' performance.

The research highlights specific examples, including classic moral dilemmas like the trolley problem, to illustrate the differences in responses generated by LLMs compared to human moral intuitions. The paper also discusses the implications of these findings for understanding the capabilities of LLMs in moral reasoning and emphasizes the importance of examining the potential biases and fairness within the datasets used for training and evaluation.

By comparing machine-generated responses to human judgments, the study aims to shed light on the alignment (or misalignment) between AI outputs and human ethical standards. The authors conclude by discussing the broader significance of their findings in relation to the ethical deployment of LLMs in various applications, stressing the need for ongoing research in this critical area of AI ethics.

**Strengths:**

1. Originality
Strength: The paper presents approaches by analyzing the moral judgments of large language models (LLMs) within a multilingual context. This exploration of how LLMs interact with diverse cultural perspectives on moral dilemmas is a fresh contribution to the field, providing insights into the alignment (or misalignment) between machine-generated responses and human ethical considerations.

2. Clarity
Strength: Despite some complexity, the paper effectively organizes its findings and presents them in a structured manner. Key results are illustrated through well-designed tables and figures, aiding in the communication of the main ideas. The systematic breakdown of moral dimensions allows readers to follow the analysis logically.

**Weaknesses:**

1. Originality
Weakness: While the paper attempts to explore moral judgments in a multilingual context, it does not significantly advance the discourse beyond existing literature on moral dilemmas. Many of the concepts discussed are already well-established in moral philosophy, and the paper may not provide enough innovative perspectives to stand out in a crowded field.

2. Quality
Weakness: The reliability and validity of the moral dimension classifications could be questioned. The paper may lack comprehensive validation of the measures used to assess moral judgments, which raises concerns about the robustness of the findings. Additionally, the potential biases inherent in the training data of the LLMs themselves are not sufficiently addressed, which could impact the conclusions drawn.

3. Significance
Weakness: The implications of the findings regarding the moral judgments of LLMs are not fully explored. While the paper presents data on how LLMs align with human preferences, it falls short of discussing the broader societal implications of these judgments, particularly in high-stakes contexts such as healthcare or criminal justice. This lack of contextualization may diminish the perceived significance of the research in real-world applications.

**Questions:**

What specific measures were taken to validate (worth) the 'moral dimension' classifications used in the study? (Unfortunately, I could not find 'beauty of dimension')

Is there a discussion of the limitations of the methodologies employed, and how might these limitations affect the overall conclusions?

What are the real-world implications of the findings, particularly in high-stakes areas such as healthcare or criminal justice?

You mentioned that GPT-3 worked well in terms of the rate. However, did you check the potential issues of bias, fairness, and overall value in the dataset itself?

**Details Of Ethics Concerns:**

I am not sure about the ethical clearance and the sophisticated design of the human responses regarding the issue.

---

> ### Author Response · Authors · 2024-11-25
>
> > Originality Weakness: While the paper attempts to explore moral judgments in a multilingual context, it does not significantly advance the discourse beyond existing literature on moral dilemmas. Many of the concepts discussed are already well-established in moral philosophy, and the paper may not provide enough innovative perspectives to stand out in a crowded field.
>
> Our core contribution is to evaluate and assess the alignment of LLMs in moral dilemmas to different populations of language speakers. To make these claims, we evaluated 19 LLMs in over 100 languages and compared their responses to a dataset with over 40 million human responses collected from over 200 countries. We believe we are the first to study and evaluate pluralistic alignment at such a scale. Could the reviewer please clarify what existing literature or citations they are referring to that covers similar findings or show that these results are already well established? We would be happy to discuss these findings in our related work section and compare and contrast their findings with our own. Furthermore, our work is empirical and not philosophical. While we draw on philosophical concepts (e.g., morality and ethics), our work is aimed at having an impact on the study of large language models, AI safety, and alignment.
>
> > Quality Weakness: The reliability and validity of the moral dimension classifications could be questioned. The paper may lack comprehensive validation of the measures used to assess moral judgments, which raises concerns about the robustness of the findings.
>
> We also believe the robustness of our findings is extremely important. We have taken two steps towards this goal. First, the moral dimensions studied are inherited from a large-scale social science survey (Moral Machine) published in Nature with over 2000 citations (Awad et al., 2018). Since we aim to understand multi-lingual alignment in LLMs, we decided to use a preexisting high-impact survey with measures and dimensions that have been externally shown to be reliable and valid. Second, we have performed our own robustness checks and controls (RQ5, Section 5.5) and show that the results are consistent across prompt rephrasing. Between using an existing influential and highly validated social science instrument and conducting our own robustness tests, we have carefully controlled for the robustness of our findings.
>
> > Additionally, the potential biases inherent in the training data of the LLMs themselves are not sufficiently addressed, which could impact the conclusions drawn.
>
> While we agree that it would be interesting to analyze the relationship between LLM training and model outputs, this is far beyond the scope of our study. Firstly, all LLMs studied (including the open-weight models) are not open with respect to their training data, so we’d be guessing what these models are trained with. Post-training and alignment for these models are even less well understood. Finally, understanding the link between training data and moral biases deserves a manuscript of its own and goes beyond the scientific contributions we aim to make in this work. Instead, our focus is the study of the biases and preferences these models exhibit due to their pretraining and finetuning and how those preferences vary across languages. We will make these ideas more clear in the discussion of limitations.

---

> > ### Comment · Reviewer_Q4Jg · 2024-11-25
> > **Disagree**
> >
> > Unfortunately, your initial rebuttal (as shown in the following statements) did not address my critique. My intention was to emphasize that, in my view, the larger size and diversity of the dataset cannot be considered a contribution when the credibility and safety of the data have not been thoroughly discussed prior to its collection. It is essential to demonstrate the integrity of human responses.
> >
> > "To make these claims, we evaluated 19 LLMs in over 100 languages and compared their responses to a dataset with over 40 million human responses collected from over 200 countries. We believe we are the first to study and evaluate pluralistic alignment at such a scale. Could the reviewer please clarify what existing literature or citations they are referring to that cover similar findings or demonstrate that these results are already well established?"
> >
> > On the other hand, your additional rebuttal helped me understand your perspective more clearly. Therefore, I have revised my judgment accordingly.

---

> ### Author Response · Authors · 2024-11-25
>
> > Significance Weakness: The implications of the findings regarding the moral judgments of LLMs are not fully explored. While the paper presents data on how LLMs align with human preferences, it falls short of discussing the broader societal implications of these judgments, particularly in high-stakes contexts such as healthcare or criminal justice. This lack of contextualization may diminish the perceived significance of the research in real-world applications… What are the real-world implications of the findings, particularly in high-stakes areas such as healthcare or criminal justice?
>
> Thank you for the interesting suggestion! Thinking about the implications of this work for high-stake areas such as healthcare or criminal justice is a great idea. Although our stimuli do not directly concern healthcare or criminal justice, the relatively abstract moral dilemmas we study share structural similarities with social dilemmas in these high-stakes areas. For instance, in the healthcare setting, a hospital may need to select who to treat with a limited supply of medicine or medical devices i.e., choosing who to save between two groups of people as in our stimuli. LLMs may encode language-sensitive biases concerning who should be given priority access to these resources. In the context of criminal justice, the victim's identity might lead to different judgments of fair punishment, similar to the dimensions we studied in this work. For instance, we might expect a higher punishment to a perpetrator if a child is harmed than when an adult is harmed. These biases may also be reflected in LLM responses. Thank you for the idea – we will discuss these possibilities in the future work section of our paper.
>
> > What specific measures were taken to validate (worth) the 'moral dimension' classifications used in the study? (Unfortunately, I could not find 'beauty of dimension')
>
> This is an important point that we will clarify with additional text. We have added the following paragraph to address that our work does not aim to validate the worth of any particular moral dimension, as this is far outside the scope of our empirical study on LLM alignment.
>
> “The Moral Machine dataset (Awad et al. 2018) is a _descriptive_ measure of human responses to moral dilemmas. It cannot (and should not) _prescribe_ how the system should act. However, because these systems are trained and fine-tuned with large-scale cross-cultural data, they may inherit culturally specific biases. Our focus here is to both characterize these biases and study to what extent input in different languages leads to an LLM reflecting the moral values consistent with the speakers of that language. From this perspective, a higher alignment score for a particular culture may or may not be desirable. While some point to reflecting human values in AI systems as a solution to the alignment problem, our results suggest that due to the plurality of moral values expressed, successful alignment must ultimately depend on the intent of the system developers to avoid perpetuating human preferences that are considered harmful, partial, or parochial.”
>
> > Is there a discussion of the limitations of the methodologies employed, and how might these limitations affect the overall conclusions?
>
> Yes! We discuss limitations in both Section 6 (pages 9-10) where we describe limitations (and future mitigation strategies) related to refusal rates, variations across different domains of moral reasoning, and better dialect support. Furthermore, we provide additional discussion of limitations in the Ethics Statement (pages 10-11, especially the last paragraph).
>
> > You mentioned that GPT-3 worked well in terms of the rate. However, did you check the potential issues of bias, fairness, and overall value in the dataset itself?
>
> Yes! Characterizing the variability of human preferences and biases on moral stimuli is a key goal of our work. Because our dataset contains judgments from 100s of countries, we can give a more detailed quantitative account of LLM biases than previously studied.
>
> ---
>
> We hope our answers address your concerns and can clarify the value of our work. In case you need any followup information, we are more than happy to provide more details!

---

### Author Response · Authors · 2024-12-03

Firstly, we would like to thank all reviewers for a wealth of feedback. Three out of four reviewers recommended 8 out of 10, and we believe we have addressed the key concerns of Reviewer Q4Jg directly, leading to their increased score after the rebuttal.

We are very encouraged by the large number and diversity of positive comments:

1. **Important and Novel Topic:** The paper “addresses an important new challenge” (JShf), “presents a novel dataset” (DzVX), “introduces an innovative approach” (sLsu), and is “a fresh contribution” (Q4Jg). The work “serves as both a technical guide for measuring alignment and a reminder of the complexities involved in cross-cultural moral decision-making in AI” (sLsu).

2. **Valuable Dataset:** Our Multilingual Trolley Problems (MultiTP) dataset “spans numerous languages” (JShf), “encompasses a variety of cultural perspectives” (Q4Jg), “highly useful for future studies on aligning model behavior with human ethics” (JShf), including “97,520 translated moral dilemma scenarios” (sLsu), “providing a comprehensive dataset for multilingual moral alignment” (sLsu), and presenting “a solid design with parametric variation and covers 107 languages, 6 moral dimensions, and 18 characters” (DzVX).

3. **Solid Study Design:** The “systematic evaluation of LLMs' moral reasoning across linguistic contexts” (sLsu) “demonstrates rigorous methodology” (sLsu), and “Key contributions include [...] developing methods to analyze variations in moral dilemmas, and creating frameworks for assessing cross-cultural alignment” (sLsu). “The systematic breakdown of moral dimensions allows readers to follow the analysis logically” (Q4Jg).

4. **Comprehensive Experiments:** The study conducts “a comprehensive evaluation of 19 LLMs, systematic moral dimension variations, robust prompt paraphrasing tests, and careful statistical analysis with transparency in acknowledging limitations” (sLsu). The “detailed experiments” (JShf) “prompts 19 LLMs” (DzVX) and “provided valuable insights” (JShf), with the experimental protocol being “robust” (DzVX).

5. **Valuable insights:** “The work contributes significantly to understanding cross-cultural AI alignment, challenges assumptions about language inequality, and provides essential tools for future research, with implications for creating culturally aware and fair multilingual AI systems” (sLsu).

6. **Clear and Access Writing**: We received a perfect presentation score (4/4) across all reviewers, who commented that “The paper is well-structured and clearly written” (JShf), “the paper effectively organizes its findings and presents them in a structured manner. Key results are illustrated through well-designed tables and figures” (Q4Jg), “making it easy to follow and understand the key findings.” (DzVX), and also “The paper is well-organized with clear research questions, strong visual aids, detailed methodological documentation, and effective use of examples to communicate complex findings” (sLsu).

Stemming from the thoughtful questions of the reviewers, we further validated our translation procedure. We introduced results from a new study we conducted on Amazon Mechanical Turk to evaluate translation quality, which that translation quality was high in the 44 languages we tested. We provide detailed explanations in our rebuttal and have also updated the PDF accordingly. Manual annotation shows. Quoting Reviewer DzVX’s supportive comment: “The data is translated using a machine translation engine, which is suitable for this task since it is not tied to linguistic subtleties.”

---

Moreover, as discussed in our individual replies, we expanded our manuscript addressing minor issues in response to reviewer concerns. We report our changes **C1-6** below:

- **C1:** We enhanced Sec 3.2 to highlight our differences from the Moral Machine study psychology and our unique contributions, especially corresponding to the pluralistic alignment research _(in response to Reviewers sLsu and Q4Jg)_
- **C2:** We included the multilingual prompt quality check in Sec 3.2 and Appendix D (with details for about 2 pages long) _(in response to Reviewers SLSu and JShf)_
- **C3:** We also discussed potential limitations of the language-based analysis in Section 6 _(in response to  Reviewer DzVX)_
- **C4:** We enriched the Ethics Statement to address the additional points from the reviewers _(in response to  Reviewer JShf)_
- **C5:** We provided additional visualizations of regional differences across Global West, East, and South in E.4.1 with Figures 9 to 11 _(in response to  Reviewer sLsu)_
- **C6:** We reported the performance of an important robustness test on the option bias in our LLM results in Appendix E.5. We find that, benefited from our prompt design, there is little such bias when we swap the order of the two choices in our MultiTP questions, as most models have 97-99.5% consistency rates in Table 13 _(in response to Reviewer DzVX)_

Thank you again to everyone!

---

### Meta-Review · Area_Chair_sm3j · 2024-12-20

**Metareview:**

This paper evaluates the moral alignment of LLMs with human preferences across different languages and cultures using trolley problem scenarios. A dataset called MultiTP containing 97,520 moral dilemma scenarios in 107 languages is introduced. The authors evaluate 19 LLMs across six moral dimensions (species, gender, fitness, status, age, and number of lives).

Strengths: The introduced dataset is novel and comprehensive, the evaluation involves multiple LLMs and languages. Overall it is an important contribution to understanding cross-cultural AI alignment.

Weaknesses: This paper would benefit from more discussions and analysis on multiple aspects raised by the reviewers, e.g., exploration of why different LLM families exhibit varying behaviors, and translation quality in very low-resource languages. But those are relatively minor issues and easy to solve in the final version.

The paper makes important contributions to understanding cross-cultural AI alignment through a novel dataset and comprehensive empirical evaluation. I believe it will provide valuable insights for both researchers and practitioners working on the alignment problem. I recommend to accept this paper. However, the potential ethical concerns, such as bias or fairness issues in different cultures in the dataset, as mentioned by multiple reviewers, suggest an additional ethical check of this paper, where the authors need to clearly explain and elaborate on those issues.

**Additional Comments On Reviewer Discussion:**

The discussion phase was particularly productive, with significant engagement between authors and reviewers. The authors' detailed responses and additional analysis led three reviewers to increase their original scores (from 3,6,6,6 to 5,8,8,8). The addressing of some concerns, particularly regarding translation quality and ethical implications, strengthened the paper significantly.

---

### Decision · Program_Chairs · 2025-01-22

Accept (Spotlight)